# Amikacin Coated 3D-Printed Metal Devices for Prevention of Postsurgical Infections (PSIs)

**DOI:** 10.3390/pharmaceutics17070911

**Published:** 2025-07-14

**Authors:** Chu Zhang, Ishwor Poudel, Nur Mita, Xuejia Kang, Manjusha Annaji, Seungjong Lee, Peter Panizzi, Nima Shamsaei, Oladiran Fasina, R. Jayachandra Babu, Robert D. Arnold

**Affiliations:** 1Department of Drug Discovery and Development, Harrison College of Pharmacy, Auburn University, Auburn, AL 36849, USA; czz0053@auburn.edu (C.Z.); izp0018@auburn.edu (I.P.); nzm0076@auburn.edu (N.M.); xzk0004@auburn.edu (X.K.); mza0153@auburn.edu (M.A.); prp0003@auburn.edu (P.P.); 2Faculty of Pharmacy, Mulawarman University, Samarinda 75119, Indonesia; 3Department of Mechanical Engineering, Samuel Ginn College of Engineering, Auburn University, Auburn, AL 36849, USA; szl0169@auburn.edu (S.L.); shamsaei@auburn.edu (N.S.); 4National Center for Additive Manufacturing Excellence (NCAME), Auburn University, Auburn, AL 36849, USA; 5Department of Biosystems Engineering, Samuel Ginn College of Engineering, Auburn University, Auburn, AL 36849, USA; fasinoo@auburn.edu

**Keywords:** 3D-printing technology, biomedical implants, personalized delivery, amikacin, polymeric coating

## Abstract

**Background/Objectives**: Personalized 3D-printed (3DP) metallic implants delivery systems are being explored to repair bone fractures, allowing the customization of medical implants that respond to individual patient needs, making it potentially more effective and of greater quality than mass-produced devices. However, challenges associated with postsurgical infections caused by bacterial adhesion remain a clinical issue. To address this, local antibiotic therapies are receiving extensive attention to minimize the risk of implant-related infections. This study investigated the use of amikacin (AMK), a broad-spectrum aminoglycoside antibiotic, incorporated onto 3D-printed 316L stainless steel implants using biodegradable polymer coatings of chitosan and poly lactic-co-glycolic acid (PLGA). **Methods**: This research examined different approaches to coat 3DP implants with amikacin. Various polymer-based coatings were studied to determine the optimal formulation based on the characteristics and release profile. The optimal formulation was performed on the antibacterial activity studies. **Results**: AMK-chitosan with PLGA coating implants controlled the rate of drug release for up to one month. The 3DP drug-loaded substrates demonstrated effective, concentration-dependent antibacterial activity against common infective pathogens. AMK-loaded substrates showed antimicrobial effectiveness for one week and inhibited bacteria significantly compared to the uncoated controls. **Conclusions**: This study demonstrated that 3DP metal surfaces coated with amikacin can provide customizable drug release profiles while effectively inhibiting bacterial growth. These findings highlight the potential of combining 3D printing with localized delivery strategies to prevent implant-associated infections and advance the development of personalized therapies.

## 1. Introduction

Accidents, conditions, and joint issues, such as osteoarthritis or rheumatoid arthritis, may require surgical procedures involving joint replacement or internal fixation devices [1]. Orthopedic implants are used widely to repair bone fractures through surgical immobilization, and these metallic devices provide excellent mechanical properties and biocompatibility [2]. Recent reports indicated a growing demand for orthopedic implants, with more than 3.8 million hip and knee replacement procedures performed each year in the United States [3], and the global market is expected to reach an estimated value of $79.5 billion by the year 2030 [1]. Polymers and ceramics have been utilized as orthopedic biomaterials; however, metals remain the most widely used due to their superior mechanical properties and corrosion resistance [2]. Stainless steels (SS), titanium-based alloys, and cobalt-chromium-based alloys are the three most common alloys in biomedical applications [2]. Stainless steels are among the most utilized metals in the production of orthopedic implants due to their antibacterial properties and resistance to staining [4]. Surgical grade 316L SS is gaining attention for biomedical use because it offers good mechanical strength, strong corrosion resistance, general biocompatibility, and cost-effectiveness [5,6]. Its composition and low carbon content make 316L SS suitable for reducing the corrosion risk and are appropriate for medical applications [7,8].

3D printing (3DP), also referred to as additive manufacturing (AM), fabricates 3D physical structures by successive addition of materials using a computer-aided design (CAD) model [9]. 3DP has received extensive attention in the fabrication of implants recently, which allows for the customization of medical implants tailored to individual anatomical and clinical demands, offering the potential for improved performance and higher effectiveness compared to conventional mass-produced devices [4]. Moreover, the ability to precisely control geometry, porosity, and surface features makes 3DP advantageous for enhancing biological compatibility and optimizing drug delivery capabilities.

Several well-known fabrication techniques have been explored in 3D printing, including laser powder bed fusion (L-PBF), stereolithography (SLA), inkjet printing, selective laser sintering (SLS), and bioprinting [10]. Laser powder bed fusion (L-PBF) methodology is an additive manufacturing process that uses a high-powered laser to selectively melt and fuse layers of fine metal powder to create a 3D object based on a digital model [11]. Fabrication involves spreading a thin layer of powder over the build platform, and a laser scanning the desired cross-section, melting the powder until the complete object is formed [11,12]. L-PBF enables the fabrication of highly complicated and precise components with excellent mechanical properties and structural control, making it suitable for producing biomedical implants [13]. Additive manufacturing enables customized modification of implants, allowing for precise alterations in shape, size, materials, surface roughness, weight, and drug-loading [14,15,16]. This flexibility supports the development of lightweight implants with enhanced capabilities for treating infections and inflammation.

However, post-surgical infections (PSIs) after implantation have been a complex challenge in modern medicine, which leads to increased healthcare costs, prolonged hospitalization, life-threatening complications, severe patient morbidity, and implant failure [17,18]. Implant-associated infections are mainly caused by bacterial adhesion and proliferation on the surface of the implants. Microbial adherence to the surface is followed by the aggregation of microorganisms and biofilm formation [19,20]. The surface of an implant can be susceptible to bacterial colonization, particularly by *Staphylococcus species*, which are responsible for approximately 80% of implant-associated infections in humans [21]. *Staphylococcus aureus* accounts for approximately 15.6%, and *Staphylococcus epidermidis* accounts for about 42% [21]. Bacterial contamination during or after surgery can lead to localized infections that are difficult to treat, especially when antibiotic-resistant strains are involved. Thus, a potential enhancement option is applying bioactive antimicrobial coatings to the surfaces of orthopedic implants to extend their durability by promoting biocompatibility, increasing corrosion resistance, and reducing implant failure caused by infections [21,22]. Multiple coating techniques have been investigated for their effectiveness in preventing implant-related infections, such as spin coating, electrospun coatings, electrophoretic deposition, spray coatings, lipid-based coatings, and 3DP-based coatings [23,24]. Each method offers unique surface modification benefits for improved implant performance, and the roughness of the implant surface facilitates drug deposition to promote bioactivity [24].

Antibiotics commonly used for implant coatings include vancomycin, gentamicin, amikacin, tobramycin, rifampicin, linezolid, daptomycin, and ciprofloxacin [24,25]. The selection is based on their effectiveness against pathogens like *Staphylococcus aureus* and *Staphylococcus epidermidis*. These antibiotics are often incorporated into biodegradable polymers [16]. Such drug-polymer coatings enable localized, sustained drug release directly at the implant site, preventing infection while reducing systemic side effects with local drug delivery. Our previous study demonstrated that gentamicin coating effectively prevents implant-related infections based on in vitro antimicrobial assays [7,8]. While amikacin and gentamicin are aminoglycoside antibiotics with broad antibacterial activity, amikacin offers advantages over gentamicin, particularly in its efficacy against methicillin-resistant *Staphylococcus aureus* (MRSA) [26].

Amikacin (AMK) is a third-generation semi-synthetic aminoglycoside antibiotic derived from kanamycin A with the side chain modification (Figure 1A) [27]. It is a broad-spectrum aminoglycoside antibiotic and has shown potent activity against a wide range of Gram-negative and some Gram-positive pathogens commonly implicated in PSIs [28]. The bactericidal effect of amikacin is mainly achieved by hindering the synthesis of bacterial proteins, and the drug can affect the entire process of bacterial protein synthesis and increase the permeability of cell membranes [28,29].

Polymers used in implant coatings serve as carriers for antibiotics to control drug release and improve biocompatibility. Common biodegradable polymers include poly lactic acid (PLA), poly glycolic acid (PGA), poly lactic-co-glycolic acid (PLGA), and polycaprolactone (PCL) [30]. These polymers gradually degrade in the body and release therapeutic bioactive agents over time. Natural polymers like chitosan, gelatin, and alginate are also used for their biocompatibility and antimicrobial properties [16,30]. Chitosan is a natural and biodegradable polysaccharide derived from chitin (Figure 1B), known for its non-toxic, biocompatible, biodegradable, and antimicrobial properties [31,32]. PLGA is a synthetic biodegradable copolymer widely used in medical applications for its biocompatibility and degradation [33]. PLGA can provide sustained release of bioactive molecules at the desired rate to reduce infections around implants [34].

In the present study, we investigated the efficacy of amikacin-coated 3D-printed metal devices in mitigating postsurgical infections by combining advanced manufacturing technology with localized antibiotic therapy. Three-dimensional printing technique offers a promising platform for personalized drug-loaded coatings with designed shapes and controlled release kinetics. This work aims to develop an amikacin-loaded chitosan coating, further encapsulated with a PLGA layer onto the metal implants, for application on postsurgical infections. The formulation was evaluated for its physicochemical characteristics, drug release behavior, kinetic models, and antimicrobial performance against pathogens. The results will support the novel coated implants as a promising approach to promote surface biocompatibility, reduce the risk of postsurgical infections, and enhance surgical outcomes.

## 2. Materials and Methods

### 2.1. Materials

Amikacin base was purchased from Cayman Chemical Company (Ann Arbor, MI, USA). PLGA (Poly-d, l-lactide-*co*-glycolide) Resomer^®^ RG 756 S (lactide: glycolide 75:25, mw 76 kDa–115 kDa), chitosan with low molecular weight (LMW; mw < 100 kDa, viscosity 20 cps) and medium molecular weight (MMW; mw 100 kDa–1000 kDa, viscosity 200 cps) were obtained from Sigma Aldrich (St. Louis, MO, USA). 1-Fluoro-2,4-dinitrobenzene, purity > 99%, was purchased from Alfa Aesar (Ward Hill, MA, USA). PBS tablets, acetic acid glacial, acetone, and analytical HPLC grade acetonitrile and methanol were acquired from VWR chemicals (Radnor, PA, USA). Amikacin minimal inhibitory concentration (MIC) test strips were purchased from Liofilchem, Inc. (Waltham, MA, USA). Tryptic soy broth (TSB) was acquired from Ward’s Science (West Henrietta, NY, USA). Tryptic soy agar (TSA) plates were procured from Hardy Diagnostics (Santa Maria, CA, USA). Ultrapure water (>18.3 mΩ) was obtained with the PICOPURE^®^ 3 water purification system (Durham, NC, USA).

### 2.2. Implant Prototypes Fabrication

The Renishaw AM250 laser powder bed printer, a selective laser melting (SLM) system, uses a fused deposition method [35]. This 3D printer was used to produce 316L stainless-steel (SS) metal implant prototypes with a layer-by-layer additive manufacturing technique, as shown in Figure 2. The fabrication process was followed based on the previously published article [8]. The design of each specimen was a square block with a size of 2 cm × 2 cm × 0.3 cm using a Computer-Aided Design (CAD) file. Before the fabrication process, the plate was heated to 80 °C and then cooled down with a high cooling rate. The build orientation and the build platform were formed in diagonal lines. The whole process was conducted under argon gas conditions to avoid oxidation. The fabrication process was carried out using the default parameters for 316L stainless steel, with a layer thickness of 50 mm, a hatching distance of 110 mm, a laser power of 200 W, a laser point spacing of 60 mm, and a laser exposure time of 80 ms. The settings were adjusted to provide a consistent roughness range of the implant surface. And the meander pattern of the laser scan strategy could achieve the same shape on each layer.

### 2.3. Preparation of Hydrogel for 3DP-Based Coating

Low-molecular-weight chitosan (2%, *w*/*v*) and medium-molecular-weight chitosan (2%, *w*/*v*) were dissolved in a 1% (*v*/*v*) acetic acid solution (pH 2.9), respectively. The mixture was stirred continuously using a magnetic stirrer until all particles were thoroughly dispersed and evenly distributed before printing. After that, amikacin base was added to different molecular weights of chitosan and sonicated until all drugs were dissolved entirely in the hydrogel.

### 2.4. Implant Coating with 3D-Printing and Drop Casting Technique

Before coating, all metal specimens were autoclaved and exposed to UV radiation for sterilization. Figure 2 briefly represents the coating process of different polymeric layers. The amikacin-chitosan coating was accomplished with a single layer through a syringe-extrusion 3D printing technique with a BioX 3D printer (Cellink LLC, Gothenburg, Sweden), and implants were allowed to dry overnight in a 30 °C oven. The printing model was designed using computer-aided design (CAD) software (SolidWorks 2021, version 29.0, Dassault Systèmes SolidWorks Corp., Waltham, MA, USA) and was 1.5 cm × 1.5 cm × 0.3 cm in size. The uniform coating was conducted by following printing process parameters (Detailed in Table 1): syringe pump of 2.5 mL, nozzle diameter of 0.25 mm, printing speed of 2 mm/s, extrusion rate of 1 μL/s, retract volume of 10 μL, infill pattern of grid, infill density of 98%, layer height (80%) of 0.25 mm. After drying thoroughly, 50 mg/mL of PLGA was dissolved in acetone, and 500 μL of PLGA was deposited onto the implant surface with another layer using a drop-casting technique with the pipette. Then the coated implants were stored in sterile containers and dried in the oven for 12 h. All implant coating processes were performed under sterile conditions with UV exposure in the printer.

### 2.5. Characterizations of Coated Implants

#### 2.5.1. Surface Morphology

The surface morphology of coated and uncoated 316L SS metal implants was characterized with a Keyence VHX-6000 optical and digital microscope (Keyence, Osaka, Japan). A 20 mg blend with 10% (*w*/*w*) drug loading implants was selected for the experiments. The top view of the surface was observed at 100× magnification with different coated metal implants.

#### 2.5.2. Roughness and Thickness Measurement

Roughness and thickness measurement were performed using a Keyence-VHX 600 microscope with the 3D-stitched function. The 3D image of the coated metal implant, and line roughness measurement were achieved with the 3D-stitching function at 500× magnification. Multiple parallel lines were profiled for each specimen to reduce statistical impact, and each line was profiled 3 mm in the same direction as the 3D-printed implant’s build orientation to obtain a roughness profile (Figure 3A). The profiled region was investigated by horizontal lines to generate the arithmetical mean deviation (Ra value) and the maximum height of the profile (Rz value). Uncoated and different coated batches were measured, and random samples were selected from each batch in triplicate. The thickness of polymeric layers was measured with the side view of the coated implant surface at 100× magnification (Figure 3B). The line measurements of the coating surface estimated the average thickness.

#### 2.5.3. Viscosity Assessment

The apparent viscosity of different molecular weight chitosan hydrogels loaded with 10% (*w*/*w*) drug samples was measured using a Brookfield DV-II+ Pro digital viscometer (AMETEK Brookfield, Middleboro, MA, USA). The viscosity values were measured at an increasing test speed with the spindle of CPA-40Z (AMETEK Brookfield, Middleboro, MA, USA) at room temperature (25 °C). The viscometer was calibrated and mechanically set before sample measurement. Each sample was measured in triplicate at a specific speed. The variation in viscosity (cP) was then plotted versus shear rate (s^−1^) with the different polymer hydrogel samples.

#### 2.5.4. Coating Efficiency

In this research, coating efficiency indicates efficiency according to the amount gained after drug and polymer deposition on the metal implants. The implant surface was coated with drug and polymer by the 3D printer, dried thoroughly, weighed for the gross weight, and then compared to the expected drug and polymer deposition amount. The coating efficiency (%) was estimated according to the following equation:(1)$$\mathrm{coating}\;\mathrm{efficiency}\;(\%)=\frac{\mathrm{Dried}\;\mathrm{coated}\;\mathrm{implant}\;\mathrm{weight}\;\left(w_{2}\right)-\mathrm{Initial}\;\mathrm{uncoated}\;\mathrm{implant}\;\mathrm{weight}\;\left(w_{1}\right)}{\mathrm{Expected}\;\mathrm{drug}\;\mathrm{and}\;\mathrm{polymer}\;\mathrm{deposition}\;\mathrm{amount}\;\left(w_{3}\right)}\;\times \;100$$

#### 2.5.5. Drug Content Uniformity

Coated implants were soaked completely in film digestion media of 1% (*v*/*v*) acetic acid and sonicated at room temperature until completely dissolved. After the whole coated layer was detached from the implant surface and dissolved in the media solution, the drug amount present in the media was detected by HPLC analysis, as detailed in Section 2.7 below. A 10% variability of the desired theoretical drug content was accepted for further study, as such deviations are acceptable in formulation processes without significantly affecting efficacy or safety.

#### 2.5.6. Differential Scanning Calorimetry (DSC)

Thermal stability analysis was conducted using a differential scanning calorimeter (TA Instruments, Model Q200, New Castle, DE, USA). The analysis involved a two-stage heating process from 10 to 350 °C at a rate of 10 °C per minute under an inert nitrogen atmosphere (20 psi). Each sample, weighing between 6 and 7 mg, was measured with a calibrated balance and sealed in aluminum pans (DSC Consumables Inc., Austin, MN, USA). The polymeric layer was carefully removed using a sterile stainless-steel blade to collect the sample for analysis. The resulting thermograms were interpreted using Universal Analysis 2000 software (TA Instruments, New Castle, DE, USA) to determine melting point, glass transition temperature, and crystalline behaviors.

### 2.6. In Vitro Drug Release

The in vitro drug release profiles were assessed by incubating drug-loaded implants at 37 °C for one month. All implants were immersed in release media consisting of 3 mL of phosphate-buffered saline (PBS) at pH 7.4. The release study was performed in a 6-well culture plate, and the implant was positioned with the drug-coated surface facing downward, in direct contact with the stainless-steel mesh. The top of each well was covered with parafilm tightly, and the lid was sealed with parafilm again to reduce media evaporation in the 37 °C incubator. All batches were prepared in triplicate. At each pre-established time point (2, 4, 8, 12, 24, 48, and 72 h), release media were transferred into a 5 mL tube, and 3 mL of fresh media was transferred into each well. The sample suspension was centrifuged in an IEC Clinical Centrifuge (Needham Heights, MA, USA) for 20 min at 1790 g at room temperature.

The release profile was determined directly by quantifying the amount of drug present in the supernatant. The supernatant was then mixed with a derivatization reagent and diluted with methanol [36]. The drug content within the reaction system was determined using a validated HPLC method described below. Samples were run in triplicate, and results were reported as a percentage (%) of the drug cumulative release. To detect residual drug content in the coated layers, when the drug release study was completed, all implants were separately transferred to 30 mL borosilicate glass vials capped with polytetrafluoroethylene (PTFE) lined polyethylene caps that were filled with film digestion media of 1% (*v*/*v*) acetic acid and sonicated at room temperature overnight. Similarly, the drug amount in the digestion media was also quantified with an HPLC analysis. This study was conducted using various drug loading and polymer composition coatings on metal implants. The initial burst release phase was evaluated for linearity using several mathematical kinetic models, including a zero-order, first-order, Higuchi, Hixson-Crowell, and Korsmeyer-Peppas [37].

### 2.7. HPLC Quantification of Amikacin

#### 2.7.1. Method Development

A high-pressure liquid chromatography (HPLC) with visible absorbance (UV detector) analysis method was developed and validated to quantify the amount of drug coating the surface of the implants. Amikacin is an aminoglycoside that lacks a chromophore and requires derivatization for its UV/VIS detection (Figure 1C). The derivatization system composition and volume were modified and optimized based on the previously published paper [36] and the instrument. The obtained mixture was then heated at 90 °C in the water bath on the heating plate for 10 min, cooled to room temperature, and a 10 μL volume was injected into the HPLC. The standard stock solution of amikacin was scanned at 700–300 nm using a UV/VIS Spectrophotometer (V-630, JASCO, Tokyo, Japan) to determine the maximum wavelength (λ_max_) for absorption (Appendix A). After λ_max_ determination, analytical method variables such as mobile phase composition, column selection, column temperature, injection volume, running time, running condition, and flow rate were optimized.

#### 2.7.2. Instrumentation and Chromatographic Conditions

Liquid chromatography was performed on a Waters e2695 separations module, and the detection was conducted on a Waters 2998 photodiode array (PDA) detector. Data acquisition and quantification were performed using Empower^®^ 3 software (Waters Corporation, Milford, MA, USA). As shown in Appendix A, the chromatographic separation was performed using a reverse-phase Phenomenex Luna 5μ C_18_ column (250 mm × 4.60 mm, i.d. 5 μm). The sample was eluted for separation with an isocratic mobile phase comprising acetonitrile-water-acetic acid (45:55:0.1, *v*/*v*/*v*) at a flow rate of 1 mL/min. The column temperature was controlled at 45 °C, and the autosampler temperature was maintained at 25 °C. The injection volume was 10 μL, and the detection was carried out at 323 nm using a PDA detector.

#### 2.7.3. Preparation of Standard Solutions and Sample Solutions

Amikacin stock solutions were prepared in deionized water at 1 mg/mL and 200 μg/mL concentrations. Stock solutions of the analyte were stored in screw-cap amber vials at −20 °C until use. To set up the calibration curves, the standard stock solution of the analyte was serially diluted with PBS to generate working standard solutions having a concentration of 0.5, 1, 5, 10, 20, 40, 80, and 100 μg/mL (Appendix A). After collection, 100 μL of the release sample was added with 700 μL of methanol, 100 μL of 5 mg/mL NaOH, and 100 μL of derivatizing agent (180 mg/mL 1-fluoro-2,4-dinitrobenzene (FDNB)). All mixtures were heated at 90 °C for 10 min, then cooled to room temperature and injected into the HPLC.

#### 2.7.4. Method Validation

Method validation was identified following the U.S. Food and Drug Administration (FDA) Bioanalytical Method Validation Guidance for Industry [38], described briefly below.

##### Linearity and Sensitivity

The linearity was evaluated by plotting eight calibration standards in the range of 0.5–100 μg/mL versus their responses in the peak area. The calibration curve was fitted by linear regression for the concentration-peak area ratio relationship. The curve was acceptable when the correlation coefficient (R^2^) was 0.99 or better and calibration standards had accurate values within the ± 15% range.

The sensitivity was assessed by analyzing six replicates of individual calibrators, and the mean concentrations were used to set up the standard curve. The LLOQ was applied as the lowest calibration standard. The limit of detection (LOD) and the lower limit of quantification (LLOQ) of the drug were determined experimentally by calculating the signal-to-noise ratio (S/N, i.e., 3.3 for LOD and 10 for LLOQ), using the following equations designated by the International Conference on Harmonization (ICH):LOD = 3.3 × *σ*/*S*
(2)LOQ = 10 × *σ*/*S*
(3)
where *σ* = the standard deviation of the response of the curve. *S* = slope of the calibration curve.

##### Accuracy and Precision

Accuracy and precision were determined by analyzing three different concentrations of quality control (QC) samples (low, medium, high) on a single day (intra-day) and three consecutive days (inter-day). Each standard solution concentration was analyzed in six replicates under the same experimental conditions. Accuracy was expressed as the percentage recovery by comparing the obtained and nominal values. Intra- and inter-day precision were expressed as relative standard deviation (R.S.D.).

##### Robustness

The robustness of the method was assessed by the analysis of slight changes in the mobile phase composition, column temperature, and flow rate. Their effects on the recovery and repeatability of the drug were investigated to study the method’s robustness.

### 2.8. Antimicrobial Study

Four bacterial isolates were selected for the antimicrobial study. These included *Staphylococcus aureus* (Tager 104 strain) from Diagnostica Stago, Parsippany, NJ; methicillin-resistant *Staphylococcus aureus* (USA 300 JE2 strain NR-46543) from BEI Resources, Manassas, VA; *Staphylococcus epidermidis* (Xen 43 strain) from Revvity, Waltham, MA; and *Escherichia coli* (Xen 14 strain) from Revvity, Waltham, MA, USA, Prior to experiments, bacterial strains were incubated in 100 mL of tryptic soy broth separately, shaking at a constant 150 rpm at 37 °C for 24 h in the MaxQ 5000 Floor-Model Shaker (Thermo Scientific, Waltham, MA, USA). After incubation, OD values were measured with the spectrophotometer at a wavelength of 600 nm to estimate the optical density. A suspension of the freshly pure culture was spread evenly with a sterile pipette over the face of the aseptic tryptic soy agar plate. An acceptable inoculum should give approximately 1–2 × 10^8^ CFU/mL [39].

#### 2.8.1. Minimum Inhibitory Concentrations (MICs) Determination

Before conducting antibacterial efficacy studies, an MIC value was determined against various bacterial strains. MIC indicates the level of susceptibility or resistance of bacterial strains to a specific antibiotic in vitro [40]. 400 μL of corresponding isolate broth culture was pipetted over the entire sterile agar surface, and the plate was rotated approximately 60 degrees to ensure an even distribution of inoculum on the surface. The agar plate was placed in the sterile hood until the inoculum was fully absorbed, and the surface was completely dry for the next step. An amikacin MIC test strip (Liofilchem, Inc., Waltham, MA, USA) was placed on the agar surface with the scale facing down and pressed with a sterile forceps to ensure complete contact between the antibiotic gradient strip and the agar surface. The strip could not be moved after application, or it would affect the results of the MIC values. The plate was then sealed with parafilm and incubated overnight in an inverted position in a 37 °C incubator. The following day, MIC values were determined at the point where the inhibition ellipse intersected the strip, representing the corresponding drug concentration in μg/mL.

#### 2.8.2. Antibacterial Efficacy Study

The well diffusion technique was used to study amikacin-coated implants’ antibacterial efficacy against various isolates. The drug loading amount within the implant coating was determined based on the previous MIC determination for each strain. Figure 4 shows the schematic design of an antimicrobial efficacy study. Three batches of implant groups were selected in this study: a group with uncoated implants, a group coated with polymer, and a group coated with 10% (*w*/*w*) drug and polymer. 3D-printed 316L SS implant (2 cm × 2 cm × 0.3 cm) was applied to the center of the agar plate. Each agar plate was sealed with parafilm and incubated for 24 h. When an antimicrobial agent leached from the object into the agar and exerted a growth-inhibiting effect, a clear zone appeared around the test metal implant. The zone of inhibition (ZOI) was visualized with 24419/B IVIS LUMINA II Imaging System (Caliper Life Sciences, Hopkinton, MA, USA). Diameters were measured from the upper surface of the agar plate, and the inhibition area was calculated for the antibacterial efficacy study.

#### 2.8.3. Antibacterial Longevity Study

An antibacterial longevity study was performed based on the results of the antibacterial efficacy study. Two optimal isolates were selected for further antimicrobial experiments. In this study, metal implants were coated with different drug loadings to compare the zone of inhibition for 7 days, and each group was conducted in triplicate. After every 24 h incubation, the implant was removed to a freshly inoculated agar plate daily for another day’s incubation. The diameters of each plate were identified once the implants were removed. The 7-day zone of inhibition would be plotted as a function of time with mean ± standard error of the mean.

### 2.9. Statistical Analysis

All the experiments were carried out in triplicate, and all data were expressed as mean ± standard error of the mean (SEM). Mean separation and significance were analyzed using the GraphPad Prism 8 (GraphPad Software, San Diego, CA, USA) software package. One-way ANOVA was used to evaluate the level of significance, and *p*-values are indicated according to significance (* *p* < 0.05, ** *p* < 0.01, *** *p* < 0.001, and **** *p* < 0.0001).

## 3. Results

### 3.1. Coating Technique Optimization

Figure 2 shows the schematic illustration for printing and depositing the polymeric coats onto the 316L SS metal implant. The substrate surface was coated with an amikacin-chitosan mixture with a 3D printer using a customized syringe-extrusion technique under sterile conditions. The parameters of coating and printing in the preparation of amikacin-coated implants were optimized according to the consistent uniformity of the coating product. The 3D-printing model was designed with Cellink HeartWare 2.4.1 (Cellink LLC, Gothenburg, Sweden), and a 1.5 cm ×1.5 cm model was created based on the substrate size with 2.0 cm × 2.0 cm. As discussed in Table 1, printing and coating parameters were generated for drug deposition onto the metallic implant. A 10% (*w*/*w*) drug was loaded into a 2% (*w*/*v*) chitosan solution, which had been prepared in advance using 1% (*v*/*v*) acetic acid. A sterile syringe filled to 2.3 mL was optimally adjusted to the syringe-extrusion printhead under instrumental settings. A nozzle with an internal diameter of 0.25 mm was selected to ensure the hydrogel passes through the nozzle without clumping and clogging. The final printing volume was 457 μL at the extrusion rate of 1 μL/s, and a 10 μL retract volume could withdraw the syringe and stop the extrusion process. The layer was printed as a grid pattern with 98% infill density. After the AMK-chitosan hydrogel coating layer dried, 500 μL of PLGA (50 mg/mL) was coated onto the implant surface using the drop casting technique with a pipette. This upper coating layer should completely cover the underlying layer of the metal implant surface to ensure encapsulation.

### 3.2. Physical Characterizations of Coated Implants

#### 3.2.1. Surface Morphology Evaluation

Figure 5 shows the microscopic surface morphology of uncoated and coated implants under the Keyence VHX-6000 optical and digital microscope. The uncoated implant surface exhibited protrusions and grooves on the substrate. All coating implant surfaces represent smooth adherence with complete cover on the rough stainless-steel substrates without visible drug precipitation under optical microscopy. In addition, implants with PLGA coating displayed much smoother and more uniform deposition patterns with LMW or MMW chitosan coats. All coating morphology images clearly show that the implant surface is fully covered with polymeric layers onto concave grooves, due to the spreading behavior effect of chitosan.

#### 3.2.2. Roughness and Thickness Measurement

The roughness was measured from the top view of the implant surface with line roughness measurement (Figure 3A). The roughness values, represented by Ra and Rz values, were plotted with uncoated and various coated batches in Figure 6A. The roughness of the surface with different batches of formulations decreased significantly (*p* < 0.05) because of coating with polymers on the rough metal surface. Batches with LMW chitosan coating blends presented a more significant decrease (*p* < 0.0001) in average roughness (Ra values) compared to the MMW chitosan coating batches, and it was considered as one of the selection criteria in the further antimicrobial study.

After PLGA coating, the thickness of the polymeric layers increased from ~290 to ~440 μm with LMW chitosan coats and increased from ~320 to ~470 μm with MMW chitosan coats, respectively (Figure 6B).

#### 3.2.3. Viscosity Assessment

The variation in viscosity versus shear rate for different chitosan hydrogel samples at room temperature is plotted and presented in Figure 7. With the spindle size of CPA-40Z, an apparent trend of these profile curves is that the viscosity of samples decreases sharply with an increase in the shear rate. The viscosity achieves a minimum level. This investigation indicates that chitosan hydrogels behave as a pseudoplastic material; this phenomenon is also known as “shear thinning”. In addition, the profile of MMW chitosan hydrogel samples displays a higher viscosity value at a constant shear rate compared to LMW chitosan hydrogel samples under the same spindle geometry and temperature conditions.

#### 3.2.4. Coating Efficiency and Content Uniformity of Coated Implants

The coating efficiency of LMW and MMW chitosan coating batches was 82.1 ± 0.6 and 84.8 ± 4.6%, respectively. And drug content uniformity for those LMW and MMW chitosan coating batches was 92.4 ± 4.6 and 98.2 ± 4.7%, respectively. Both batches showed less than 10% variability of the desired theoretical drug content, indicating they were accepted for further study.

#### 3.2.5. Thermal Analysis of Polymeric Layer

As described in Figure 8, the DSC thermogram of pure amikacin showed a sharp endothermic peak of melting point at 246 °C, similar to a previous study mentioned [41]. The DSC analysis of both LMW and MMW chitosan exhibited an endothermic peak around 108–109 °C, corresponding to the melting point, followed by an exothermic peak at 307 °C, suggesting the thermal degradation of chitosan. The glass transition temperature for PLGA was observed near 56 °C, which agrees with the literature-reported range of 40 °C to 60 °C [42,43], and displayed a sign of degradation at approximately 317 °C. The absence of significant changes in the physical mixture demonstrates that no physicochemical interaction occurred between the drug and the polymers. The coating layer of AMK-LMW-PLGA and AMK-MMW-PLGA presented broad endothermic peaks at 213 °C and 218 °C, respectively, indicating that amikacin exists in an amorphous state, which implies that it is well dispersed within the chitosan matrix.

### 3.3. In Vitro Release of Amikacin

Figure 9 shows the cumulative drug release of amikacin in PBS (pH 7.4) at 37 °C with different implant surface coatings. PBS samples were collected at each estimated time point, and then an HPLC with a UV detector was used to measure the absorbance of the derivatized sample at a wavelength of 323 nm. As depicted in Figure 9A, the release profiles with multiple drug loadings presented similar release patterns. All the profile curves showed biphasic release kinetics with a burst release during the initial 12 h, followed by a sustained release until 96 h. 2.5, 5, and 10% (*w*/*w*) drug loading batches showed a cumulative release of around 74, 83, and 90%, respectively. After a burst release phase, a relatively steady state release profile was achieved, and almost all the incorporated amikacin (>97%) was released over time. Meanwhile, the higher the amount of drug loaded in chitosan, the higher the elution value of cumulative release at the burst release phase was observed. Figure 9B displays a dependency on the chitosan molecular weight for drug cumulative release. 10% (*w*/*w*) drug was loaded in LMW and MMW chitosan separately, and both showed a burst release in the first 12 h and 8 h, respectively. Then, a release at a much slower rate was observed after the initial burst. However, at the burst release phase, batches with LMW chitosan coating eluted more (~90%) than the MMW chitosan-coated groups (~70%). In contrast, MMW chitosan batches could release for a longer period, up to 10 days, for complete release.

Figure 9C depicts drug elution profiles of different drug percentage loading with PLGA coating. All polymeric coatings with 5 and 10% (*w*/*w*) drug loadings exhibited an initial burst release within the first 7 days, followed by a sustained release phase with diffusion through the polymeric layers for one month. During the burst release phase, 10% (*w*/*w*) drug-loaded coating presented a higher cumulative release than 5% (*w*/*w*) formulation batches, indicating a dependency on the amikacin concentration of the release pattern as desired (Appendix A). In addition, 10% (*w*/*w*) AMK-LWM-PLGA batches provided an optimal release profile for drug delivery, with an initial burst release (~50%) followed by a prolonged release pattern. More than 90% of incorporated amikacin was released completely within one month.

The predicted release pattern is illustrated schematically in Figure 9E. The amikacin coatings exhibit an initial burst release upon PLGA erosion, followed by a controlled and sustained release mediated by drug diffusion and gradual erosion of the polymeric layers.

The mechanism of different release patterns combined with multiple kinetic models is presented in Table 2. The initial burst release pattern was estimated to fit with various kinetics models. The value of the regression coefficient, R^2^, close to 1 represents that the kinetic model is fitting the mechanism of drug elution. The release data of LMW chitosan coating fitted better with the zero-order equation, with R^2^ of 0.9825 for 5% (*w*/*w*) and 0.973 for 10% (*w*/*w*) drug-loaded batch, respectively, which indicates that the constant drug release rate is independent of drug amount or concentration in the delivery system. For the release mechanism of MMW chitosan batches, the data followed the Hixson-Crowell model with R^2^ of 0.9936 and 0.9964 for 5% (*w*/*w*) and 10% (*w*/*w*) drug loading implants, respectively. The drug release profile followed the Hixson–Crowell model, indicating that the release rate was influenced by changes in the surface area and shape of the polymer matrix due to erosion and dissolution over time. In addition, the MMW chitosan groups release profiles also fitted well with the Korsmeyer–Peppas model, with a higher *R*^2^ value. The value of n predicts drug release mechanism. Here, an n value between 0.45 and 0.89 suggests an anomalous (non-Fickian) transport mechanism involving a combination of diffusion and matrix relaxation.

### 3.4. Development and Validation of HPLC Method for Amikacin Analysis

Amikacin required derivatization before HPLC-UV/VIS analysis due to the lack of a chromophore for UV light detection (Figure 1C). Here, a modified HPLC method was developed and validated to quantify the drug amount on the surface of implants. The optimized method details are summarized in Appendix A, and Appendix A illustrates the UV/VIS absorption spectrum of amikacin after derivatization with FDNB reagent, indicating maximum absorption wavelength at 323 nm. The chromatogram of a derivatized standard solution for amikacin (80 μg/mL) is shown in Appendix A. The retention time of amikacin derivatives was about 15 min, and the targeted compound peak was well-defined and symmetrical.

A calibration curve ranging from 0.5 to 100 μg/mL (Appendix A) was plotted versus responses (peak area). The linearity and sensitivity of the proposed HPLC method are discussed in Appendix A. The *R*^2^ value was 0.999, and the *p*-value was found to be less than 0.0001, confirming the curve was linear and acceptable. LOD was determined to be 0.025 μg/mL, and LLOQ was established at 0.5 μg/mL based on the signal-to-noise ratio. These results are adequate for the application of a quantitative analysis of amikacin.

Appendix A describe the accuracy, intra- and inter-day precision of the proposed method. The recovery ranged between 99.9% and 103.5%, with R.S.D. being less than 15%, suggesting the methods are repeatable and reliable. The R.S.D. of repeatability and intermediate precision were found to be 3.98–11.79%, which are considered acceptable. The obtained analytical values demonstrated that the derivatization procedure and detection conditions are feasible and reproducible. The changes and results of quality control samples for evaluating the robustness of the HPLC method are shown in Appendix A. Though the conditions were altered purposefully, the peaks shape still retained sharp and symmetrical (Figures not shown). The satisfactory recovery of around 100% proved that the method was robust and had no significant effect on these changes.

### 3.5. Antimicrobial Study

A preliminary antibiotic sensitivity assay against pathogenic microorganisms was carried out in vitro with four bacterial strains. Figure 10A shows a distinct zone of inhibition in each agar plate, confirming that all bacterial strains were susceptible to amikacin, including methicillin-resistant *Staphylococcus aureus* (MRSA), which is more challenging to treat. Both *S. aureus* and *S. epidermidis* had a MIC value of 4 μg/mL, indicating amikacin potency against *Staphylococcus*. Remarkably, amikacin presented potent antibacterial activity against MRSA with an MIC value of 10 μg/mL. In addition, it showed a decrease in MIC (2 μg/mL) in the *E. coli* strain, and this data demonstrated amikacin displaying better activity against Gram-negative pathogens.

Based on previous study results, batches with LMW chitosan loaded with 10% (*w*/*w*) amikacin and PLGA coating were chosen for the further antibacterial efficacy study, and approximately 240 μg/mL drug can be released from implant surface during the first day, which is much higher than the MIC values for various strains. The amikacin on the implant surface was adequate to produce a noticeable inhibition zone surrounding the implant. Three groups of 3D-printed metal implants (uncoated implants, polymer-coated implants, and drug-polymer-coated implants) were applied onto each strain inoculated agar plate, and the antibacterial properties were investigated with a 24 h incubation. The groups of uncoated implants and only polymer-coated implants did not present any bacterial inhibition (Appendix A). As observed in Figure 10B, the group of drug and polymer-coated implants showed inhibition zones of 1.7 ± 0.2, 0.9 ± 0.2, 2.8 ± 0.3, and 2.5 ± 0.2 cm^2^ in *S. aureus*, MRSA, *S. epidermidis*, and *E. coli* inoculated agar plates, respectively. The presence of inhibition zones with drug and polymer groups confirms that amikacin exhibits potent antimicrobial activity against various bacterial species. The value of an inhibition zone in *S. epidermidis* was the highest among all tested strains. Additionally, a clear inhibition zone observed in MRSA inoculated agar plates demonstrated amikacin’s potent activity due to MRSA’s resistance to most antibiotics.

Furthermore, *S. epidermidis* and MRSA were utilized, and different concentrations of drug loading were applied to achieve prolonged antibacterial longevity studies. Similarly to previous findings, no bacterial inhibition was detected in groups with uncoated implants or those coated only with polymer (Figures not shown). Figure 11 presents the antibacterial longevity of drug-coated implants incubating in *S. epidermidis* and MRSA for 7 days. For the batches incubated in *S. epidermidis*, 5% (*w*/*w*) drug loading groups (203 μg/cm^2^) effectively inhibited the growth of bacteria during the 7-day incubation, and the zone of inhibition achieved the highest value (~2.6 cm^2^) on the third day. After 3 days of incubation, the antimicrobial activity decreased gradually. 10% (*w*/*w*) drug loading batches (406 μg/cm^2^) in *S. epidermidis* presented a similar active inhibition trend compared to the 5% (*w*/*w*) drug loading group. Still, the most effective activity occurred on the 4th day, with the inhibition zones of around 3 cm^2^. For groups with MRSA cultured agar plates, the 5% (*w*/*w*) drug-coated implants (203 μg/cm^2^) exhibited antibacterial effect (~0.9 cm^2^) on the first two days, followed by a decreasing antimicrobial activity until the 4th day and stopped inhibiting the bacteria growth after 5 days, suggesting MRSA is a cause of severe infection which is difficult to treat. Whereas the activity in MRSA was improved with 10% (*w*/*w*) drug loading groups (406 μg/cm^2^), inhibition zones were observed for 7 days, with the highest zone of inhibition approaching 1.5 cm^2^ on the third day of incubation. The antimicrobial longevity study demonstrated that the implants maintained clear zones of inhibition for up to 7 days with higher drug loading coated metal substrates.

## 4. Discussion

### 4.1. DP Coating and Physical Characteristics of Coated Implants

This study aimed to develop a polymeric coating incorporated with amikacin on a metal implant surface to prevent post-surgical inflammatory responses and improve coated implant performance. Three-dimensional printing facilitated the creation of a customized structure with a rough surface that enables effective deposition of the drug–polymer mixture for prolonged drug release. Various surface modification techniques have been explored in recent years to achieve controlled antibiotic release for localized delivery [23,44]. Previous research proved that the 3D-printing method was promising for bioactive molecules coating on the metal substrates [45]. The successful adaptation of the syringe-extrusion 3D printing method under sterile conditions highlights the technique’s suitability for medical-grade applications. The optimization of printing parameters, such as a 0.25 mm nozzle, a retract volume of 10 μL, and an extrusion rate of 1 μL/s, ensured uniform and reproducible deposition of the amikacin-chitosan hydrogel coating layer. The bioink volume was set at 2.3 mL, less than the syringe pump of 2.5 mL, because the build volume should be lower when combined with the syringe printhead that limits print box movement. Using 3DP-based coating allowed for precise modeling and design of coating dimensions consistent with the implant size [46]. Due to the hydrophobicity of PLGA, the post-printing application of a PLGA layer was achieved using a drop-casting technique to ensure antimicrobial agent encapsulation and controlled release behavior. These coating methods reflect an important design feature for dual-layer coating systems of polymers with opposite solubility. The inner hydrogel layer supports drug retention and bioactivity, while the outer hydrophobic PLGA layer offers structural integrity and delayed drug release patterns. These approaches are beneficial for orthopedic implants because early infection can be controlled, and long-term antimicrobial agent elution can be achieved.

Surface morphology analysis demonstrated a complete and uniform coating on rough metal surfaces without observable phase separation or drug crystallization. Coating adherence is important for biomedical applications [44], as inhomogeneous drug distribution may compromise therapeutic efficacy and mechanical performance. The chitosan-PLGA-coated implant surface contributed to smoother surface features and more uniform morphology, suggesting the role of PLGA as a stabilizing agent. In a similar study, Rabea et al. showed that chitosan-based biopolymer coatings exhibited excellent film-forming ability and adhesion to metallic substrates [47]. From the morphology images of all coated implants, the ability of chitosan to spread into the implant surface grooves further confirms its compatibility as a coating material. The polymeric coatings assisted in minimizing bacterial colonization and enhancing tissue integration [48].

Surface roughness was evaluated by Ra and Rz values, which represent the arithmetic mean of the absolute deviations and the maximum height from the roughness profile baseline, respectively [49]. The reduction in surface roughness upon coating indicates a smoother implant surface because the polymer fills the grooves and ridges, which is often correlated with lower bacterial adhesion and better osseointegration [45,48]. The significantly smoother surface in LMW chitosan-coated batches aligns with previous studies, where LMW chitosan offers better flowability and uniform spreading behavior [50]. Layer thickness significantly increased with the PLGA application (*p* < 0.0001), either in LMW (from ~290 to ~440 μm) or MMW (from ~320 to ~470 μm) chitosan coatings. These values are within the typical range for drug-eluting coatings and suggest the feasibility of integrating multi-layered systems without affecting mechanical properties.

The viscosity of chitosan hydrogels observed in the rheological study is ideal for extrusion-based 3D printing due to pseudoplastic behavior, ensuring precise control during deposition and reducing the risk of nozzle clogging. This shear-thinning property enables better extrusion at higher shear rates and increased post-printing stability [51]. The higher viscosity of MMW chitosan indicates a denser polymer internal structure, which could decrease the drug diffusion rate but enhance the mechanical strength of the coating film. These distinct findings are critical for adjusting drug release and kinetic behavior based on specific clinical needs. These variations could be attributed to the higher viscosity associated with higher chitosan molecular weight, which limits polymer chain mobility and promotes greater porosity [50].

The high coating efficiency of this formulation (>82%) and high drug content uniformity (<10% variability) strengthen the consistency and reproducibility of the 3D-printed coating method. Such precision is necessary for biomedical applications, ensuring consistent dosing and predictable therapeutic outcomes. The slight variation between LMW and MMW batches may be attributed to their differing viscosities and spreading behaviors, influencing the degree of drug-polymer and polymer-polymer interactions during deposition [50].

DSC analysis provided insights into the thermal behavior of the drug-polymer system. Amikacin showed an endothermic peak at 246 °C, and chitosan displayed a peak near 109 °C, consistent with previously published findings [52]. The appearance of distinct endothermic peaks for amikacin and polymers in the physical mixtures indicates that no chemical interaction occurred between amikacin and chitosan. This physical compatibility is essential for preserving drug bioactivity and ensuring formulation stability. Interestingly, the thermal stability of amikacin improved when encapsulated within the polymeric matrix of the film, as the AMK-chitosan-PLGA layer showed a shift in the broad endothermic peak. The stable polymeric layer may enhance drug stability and maintain implant efficacy during clinical application.

### 4.2. In Vitro Amikacin Release Profile with Different Molecular Weight Chitosan Coating and Drug Percentage Loading

The release profile of amikacin demonstrated that the coating method with 3D printing allows precise drug deposition to implant surfaces and controls drug release patterns, with an initial burst release followed by a sustained release. This dual-phase release is ideal for preventing early-stage infections via burst release and maintaining long-term antimicrobial effectiveness via sustained release. Higher drug loading batches (10% *w*/*w*) resulted in increased burst release (~90%) and a higher cumulative release over time compared to lower concentrations. This behavior is consistent with diffusion-based release kinetics, where a higher concentration gradient drives faster release initially [53]. For different molecular weight chitosan formulations, the faster release profiles observed in batches with LMW chitosan can be attributed to its lower polymer viscosity and less entangled polymer structure, allowing for more rapid drug diffusion. The application of PLGA coatings extended the release period to 30 days, which was essential to ensure a sufficient daily drug dose to effectively and safely inhibit bacterial growth over a defined period. This flexible dual-layer delivery system utilizes chitosan hydrophilicity for rapid initial release and PLGA hydrophobicity to prolong drug elution time. The amikacin-loaded coatings demonstrate an initial burst release due to PLGA erosion, followed by a sustained and controlled release driven by drug diffusion and the gradual degradation of the polymeric layers. In addition, LMW chitosan batches followed a zero-order release pattern (Figure 9D), indicating a constant drug release rate independent of concentration [54]. This is particularly desirable for maintaining therapeutic levels over time. Conversely, the MMW chitosan group fitted better with the Hixson–Crowell and Korsmeyer–Peppas models, suggesting that drug release was governed by erosion or shrinkage of the polymer matrix, and controlled by both diffusion and polymer relaxation [54,55]. This variability emphasizes the importance of polymer molecular weight in customizing release profiles for specific clinical needs for short-term high-dose applications or long-term maintenance delivery. The HPLC-UV method developed for amikacin quantification post-derivatization with FDNB was demonstrated to be robust, sensitive, and reproducible. This method overcomes the challenge posed by amikacin’s lack of a UV chromophore and enables accurate evaluation of drug release and stability over time.

### 4.3. Antibacterial Efficacy of Amikacin-Coated 3DP Implants

The amikacin-coated implants should possess bacteriostatic properties to inhibit bacterial adhesion, similar to previous studies [56]. This antimicrobial study demonstrates the effective antibacterial properties and sustained drug release capabilities of 3D-printed implants coated with amikacin-loaded polymer layers. The preliminary antibiotic sensitivity assay confirmed that amikacin is broadly potent against clinically relevant pathogens, even including MRSA that is hard to treat with other antibiotics, highlighting amikacin’s potential as a valuable agent in implant-related infection prevention. The MIC values observed align with previous studies reporting amikacin’s efficacy against Gram-positive and Gram-negative bacteria [57,58]. The choice of LMW chitosan loaded with 10% (*w*/*w*) amikacin and coated with PLGA proved optimal, enabling a high initial drug release (~240 μg/mL) from the implant surface during the first 24 h. This release far exceeds the MIC values for all tested strains, ensuring rapid and effective bacterial inhibition immediately after implantation, critical in preventing early-stage infections [59,60]. The absence of inhibition zones in uncoated and polymer-only coated implants confirms that the antimicrobial effect arises directly from amikacin release rather than the polymer matrix, corroborating previous findings that polymers like chitosan and PLGA primarily function as biocompatible drug carriers rather than antimicrobial agents themselves [61]. The antibacterial activity study clearly illustrated that drug-polymer coated implants created significant zones of inhibition against all tested bacteria, with the largest inhibition observed for *S. epidermidis*. The inhibition of MRSA further supports the clinical relevance of this drug delivery system, as virulent MRSA infections are difficult to treat due to multidrug resistance [59]. These results indicate that localized delivery of amikacin via polymer coatings on 3D-printed implants can achieve antimicrobial activity while potentially minimizing systemic toxicity.

An antimicrobial longevity study revealed that drug loading concentration directly affects the duration of effective bacterial inhibition. Implants with 10% (*w*/*w*) amikacin loading maintained antimicrobial activity for up to 7 days against *S. epidermidis* and MRSA, while those with 5% (*w*/*w*) loading showed a shorter efficacy period. The gradual decrease in inhibition zones over time reflects controlled drug release, consistent with the biphasic release mechanism, where chitosan enables initial burst release and PLGA sustains prolonged drug elution [62]. The extended antibacterial effect is vital for clinical applications, as infection risks persist beyond the immediate postoperative period, particularly in orthopedic implants, where biofilm formation can lead to chronic infections [17,63]. The decreased antimicrobial activity against MRSA in the 5% (*w*/*w*) loading group emphasizes the challenges caused by resistant pathogens and presents the importance of the drug dosing regimen on implant surfaces. Increased efficacy observed with the 10% (*w*/*w*) amikacin-coated implants suggests that higher local antibiotic concentrations can overcome resistance barriers without systemic side effects, as in conventional therapy [64]. These findings align with earlier research showing that local drug delivery from coated implants can prevent biofilm formation, a concern in orthopedic surgeries [8]. Moreover, the 3D printing coating technique allows for precise customization of implant geometry and drug loading, which could be tailored to individual patient demands and infection risks. This dual-layer delivery system not only presents localized and sustained antibiotic release but also potentially improves patient compliance by reducing the need for frequent systemic administration. Furthermore, maintaining consistent local drug levels may minimize the risk of antimicrobial resistance development.

## 5. Conclusions

This study successfully demonstrated that 3D-printed metal surfaces coated with amikacin could offer customizable drug release profiles, and bacterial growth was inhibited effectively. The coating technique achieved high uniformity and efficiency, while the dual-layer system provided effective antimicrobial coverage, particularly against MRSA. The sustained release profiles suggest that this approach is promising for reducing post-surgical infections and improving patient outcomes with continuous drug elution in orthopedic implant applications. Future studies should focus on in vivo evaluation of biocompatibility, osseointegration, and long-term infection prevention to translate these promising results into clinical practice. Additionally, exploring other antibiotic combinations and polymer matrices could further enhance the versatility and efficacy of drug-coated 3DP implants.

## Figures and Tables

**Figure 1 pharmaceutics-17-00911-f001:**
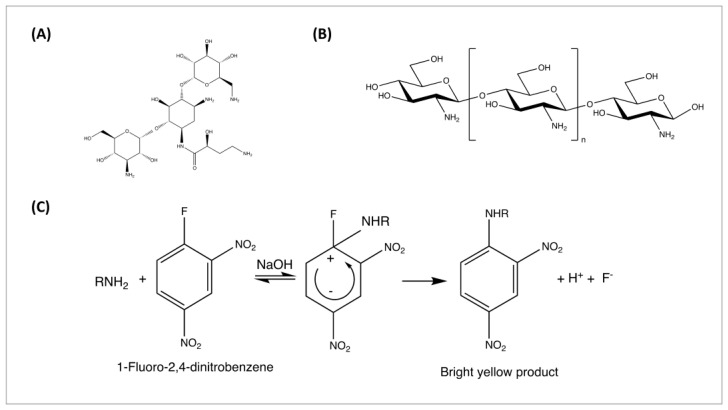
Chemical structure of (**A**) amikacin and (**B**) chitosan. (**C**) The reaction mechanism of FDNB with an aminoglycoside antibiotic.

**Figure 2 pharmaceutics-17-00911-f002:**
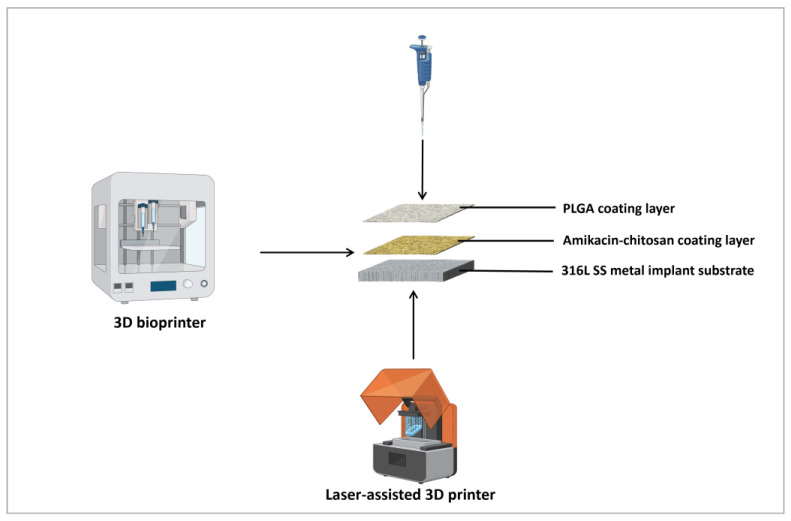
Schematic representation of 3D-printed metal implants and the coating process of polymeric layers.

**Figure 3 pharmaceutics-17-00911-f003:**
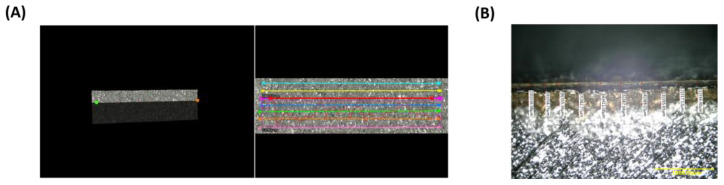
(**A**) Line roughness measurement at 500× magnification with the 3D-stitching function. (**B**) Thickness measurement using the side view of polymeric layers.

**Figure 4 pharmaceutics-17-00911-f004:**
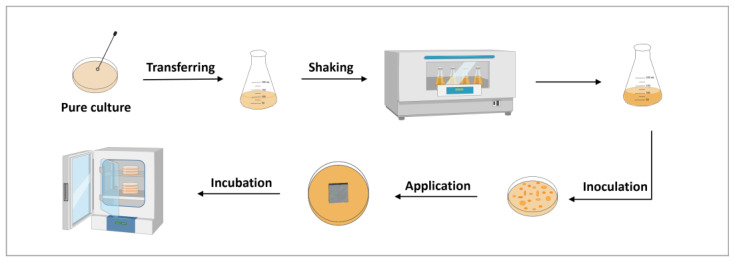
The schematic illustration of the antimicrobial efficacy study design.

**Figure 5 pharmaceutics-17-00911-f005:**
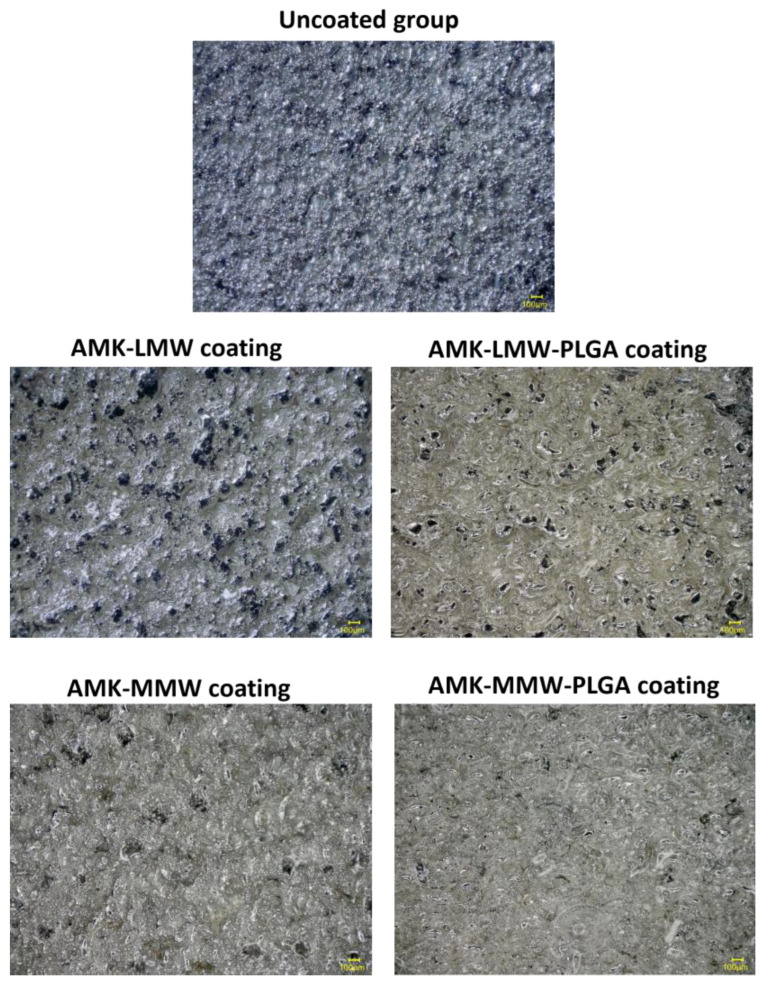
Morphological characteristics of the implant surface without and with coating (100× magnification, scale bar 100 μm).

**Figure 6 pharmaceutics-17-00911-f006:**
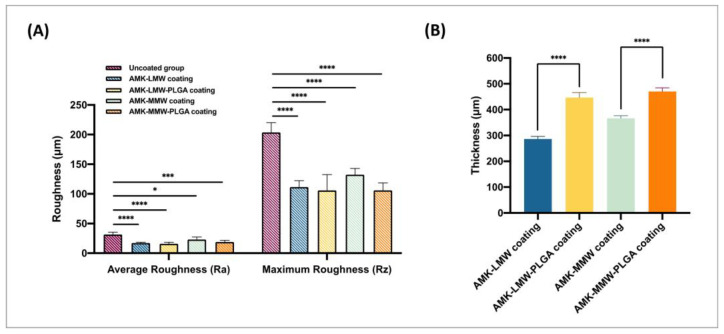
(**A**) Roughness values (Ra/Rz) with different coating formulations. (**B**) The thickness variation in different coatings on the implant surface. (Data are represented as Mean ± SEM, *n* = 3; Statistical significance was noted by *p*-value with reference to the uncoated control group. * *p* < 0.05, *** *p* < 0.001 and **** *p* < 0.0001).

**Figure 7 pharmaceutics-17-00911-f007:**
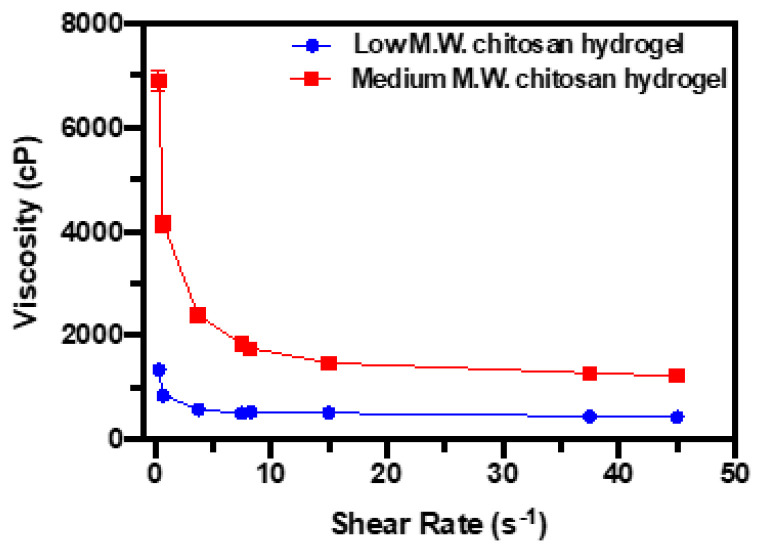
Viscosity variation in low- and medium-molecular-weight chitosan hydrogel samples at room temperature (25 °C).

**Figure 8 pharmaceutics-17-00911-f008:**
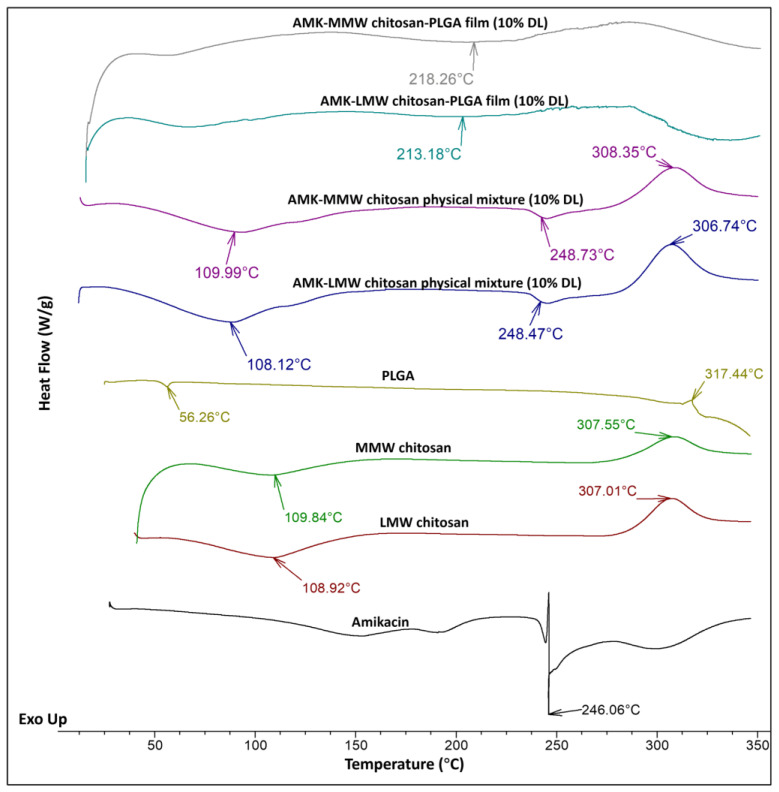
DSC thermograms of pure amikacin, LMW and MMW chitosan, PLGA, physical mixture, and coating films.

**Figure 9 pharmaceutics-17-00911-f009:**
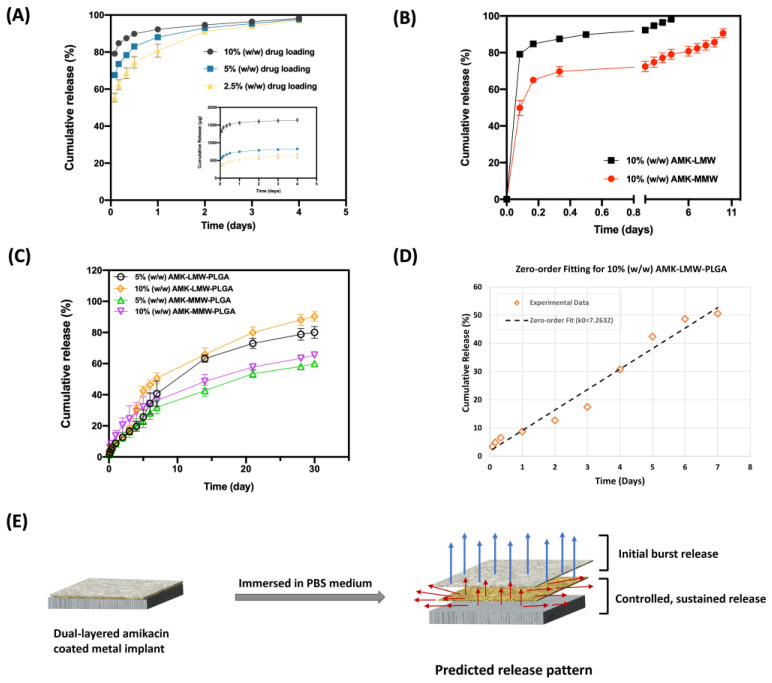
In Vitro drug release profiles of coated metal implants in PBS at 37 °C. (**A**) Profiles of different concentrations of drug loading (2.5%, 5%, 10%) in the AMK-LMW chitosan coating. (**B**) Profiles of various molecular weight chitosan loaded with only 10% (*w*/*w*) drug coating. (**C**) Cumulative release percentage of amikacin from LMW chitosan and MMW chitosan with PLGA layer coatings. (*n* = 6) (**D**) Zero-order fitting of the cumulative release profile for 10% (*w*/*w*) AMK-LMW-PLGA formulation group. (**E**) Schematic representation of predicted amikacin release pattern from dual-layered coatings.

**Figure 10 pharmaceutics-17-00911-f010:**
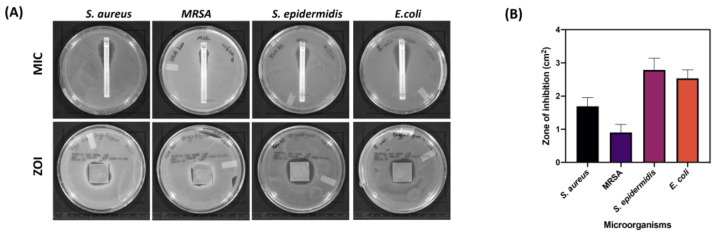
Antimicrobial study on polymeric coated implant substrates. (**A**) Macroscopic images of the minimum inhibitory concentrations (MICs) determination and observed inhibition zones (ZOI) of coated implants with four bacterial isolates. (**B**) Antimicrobial efficacy of drug-coated implants against various microorganisms for 24 h. (*n* = 3).

**Figure 11 pharmaceutics-17-00911-f011:**
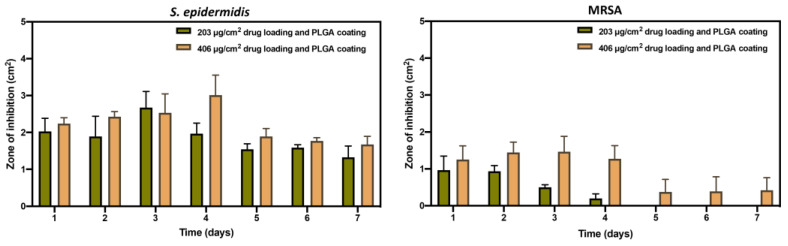
Antibacterial longevity of different drug loading and polymeric coated implants incubating in *Staphylococcus epidermidis* and methicillin-resistant *Staphylococcus aureus* for 7 days. (*n* = 3).

**Table 1 pharmaceutics-17-00911-t001:** Coating and printing parameters in the preparation of amikacin-coated implants.

Coating and Printing Parameters	Details
Concentration of chitosan	2% (*w*/*v*) LMW or MMW chitosan in 1% (*v*/*v*) acetic acid
Amikacin:chitosan ratio	1:10
Printing head	Syringe-extrusion
Nozzle internal diameter	0.25 mm (25-gauge)
Syringe pump	2.5 mL
Platform temperature	Room temperature
Printing speed	2 mm/s
Printing time	6.5 min
Printing volume	457 μL
Printing size	1.5 cm × 1.5 cm × 0.3 cm
Extrusion rate	1 μL/s
Retract volume	10 μL
Infill pattern	Grid
Infill density	98%
First layer height	0.25 mm
Concentration of PLGA	50 mg/mL in acetone
Volume of PLGA coating	500 μL/layer

**Table 2 pharmaceutics-17-00911-t002:** In Vitro release kinetics of the implants loaded with different percentages of amikacin and polymeric coatings.

Mathematical Model	Plot	Parameters Studied	5% (*w*/*w*) AMK- LMW- PLGA	10% (*w*/*w*) AMK- LMW- PLGA	5% (*w*/*w*) AMK-MMW-PLGA	10% (*w*/*w*) AMK-MMW-PLGA
Zero-order	Cumulative release (%) vs. Time	*R* ^2^	0.9825	0.973	0.9907	0.9405
		*k* _0_	5.1428	7.2632	4.1865	4.6997
First-order	Log (% cumulative release) vs. Time	*R* ^2^	0.8492	0.9381	0.8047	0.743
		*k* _1_	0.164	0.1642	0.1564	0.1274
Higuchi	Cumulative release (%) vs. Sq. root of time	*R* ^2^	0.9282	0.9018	0.9823	0.9555
		*k* _H_	14.684	20.118	12.246	14.159
Hixson-Crowell	Cube root of cumulative release (%) vs. Time	*R* ^2^	0.9725	0.9666	0.9936	0.9964
		*k* _H-C_	0.093	0.1373	0.0732	0.0854
Korsmeyer-Peppas	Log (% cumulative release) vs. Log (time)	*R* ^2^	0.9354	0.9289	0.9944	0.9819
		*n*	0.6651	0.6059	0.6549	0.5397
		*k* _K-P_	9.1285	12.574	8.4551	13.57

## Data Availability

The data presented in this study are available within the article and its Appendix A.

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
