# Peer review of "Amikacin Coated 3D-Printed Metal Devices for Prevention of Postsurgical Infections (PSIs)"

_pharmaceutics, 2025, doi:10.3390/pharmaceutics17070911_

Round 1
Reviewer 1 Report
Comments and Suggestions for Authors
Overall, this manuscript presents an interesting concept of coating 3D-printed metal with an antibiotic drug–polymer layer, followed by an outer coating of PLGA. The study demonstrates well-designed experiments and provides sufficient data to support a solid conclusion.
In lines 481–482, the thermal event observed at 56 °C is more likely an endothermic relaxation of PLGA occurring near the glass transition temperature (Tg), rather than the Tg itself. Using modulated temperature DSC could separate the endothermic relaxation from the actual glass transition.
The DSC results suggest that amikacin is in an amorphous form, which may indicate that it is well dispersed in the chitosan matrix.
Figures 9C and 9D should be combined to allow better comparison between MMW and LMW chitosan.
It is recommended to organize the discussion section into subtopics. For example:
• Coating and physical properties of the 3D-printed coating
• Amikacin release and antibacterial efficacy of the 3D coating
Author Response
Overall, this manuscript presents an interesting concept of coating 3D-printed metal with an antibiotic drug–polymer layer, followed by an outer coating of PLGA. The study demonstrates well-designed experiments and provides sufficient data to support a solid conclusion.
Thank you for your thorough review of our manuscript. We appreciate your insightful feedback and careful assessment of our work. Please find our detailed responses below, along with the revised manuscript highlighting changes in the resubmitted files.
Comment 1: In lines 481–482, the thermal event observed at 56 °C is more likely an endothermic relaxation of PLGA occurring near the glass transition temperature (Tg), rather than the Tg itself. Using modulated temperature DSC could separate the endothermic relaxation from the actual glass transition.
Response 1: Thank you for pointing this out. Modulated temperature DSC should be better at separating the endothermic relaxation from the actual glass transition. According to other research publications, PLGAs typically show a glass transition temperature ranging from 40 to 60 °C. We revised the manuscript to include additional clarification in lines 482-483.
Comment 2: The DSC results suggest that amikacin is in an amorphous form, which may indicate that it is well dispersed in the chitosan matrix.
Response 2: We appreciate the reviewer’s attention to this important detail. We modified the result descriptions in lines 487-488.
Comment 3: Figures 9C and 9D should be combined to allow better comparison between MMW and LMW chitosan.
Response 3: We agree with this comment. The combined release figure was updated in the revised manuscript in Figure 9C (line 542).
Comment 4: It is recommended to organize the discussion section into subtopics. For example:
- Coating and physical properties of the 3D-printed coating
- Amikacin release and antibacterial efficacy of the 3D coating
Response 4: Agree. We reorganized the discussion with three subtopics in the revised manuscript: in lines 634, 707, and 738.

Reviewer 2 Report
Comments and Suggestions for Authors
Dear Authors,
The manuscript presents an experimental study on amikacin coated 3D-printed implants for prevention of postsurgical infections. The work is very interesting from a practical point of view and it is written very well. The researchers employed several research techniques such as: microscopic observations, HPLC measurements, viscosity assessment, DSC analysis, release tests, antimicrobial study. The methods used in the research are clear and understandable. Undoubtedly, the authors master the approach well. However, below are questions and remarks that should be corrected before the manuscript could be consider for publication in Pharmaceutics.
- Lines 224-225. We have a different types of viscosity. Which one have you measured?
- Why did you employ the Hixon-Crowell model? This expression applies to dosage forms such as tablets. It is appropriate to describe the release profile keeping in mind the diminishing surface of the drug particles during the dissolution. I suggest to use the Korsmeyer-Peppas equation, that, apart from the release rate constant, gives the n value describing the release mechanism.
- In line 284 there was information that you used also the second-order kinetics. However, I do not see the results from this model in Table 2.
- Table 2: R2 is not a kinetic parameter. Please add the kinetic parameter such as the release rate constant for all models used.
- Can you present an example of the fitting of the release experimental data to the theoretical models?
Author Response
The manuscript presents an experimental study on amikacin coated 3D-printed implants for prevention of postsurgical infections. The work is very interesting from a practical point of view and it is written very well. The researchers employed several research techniques such as: microscopic observations, HPLC measurements, viscosity assessment, DSC analysis, release tests, antimicrobial study. The methods used in the research are clear and understandable. Undoubtedly, the authors master the approach well. However, below are questions and remarks that should be corrected before the manuscript could be consider for publication in Pharmaceutics.
Thank you for your thorough review of our manuscript. We sincerely appreciate your insightful feedback and detailed assessment of our work. Please see our comprehensive responses below, as well as the revisions highlighted in track changes within the resubmitted files.
Comment 1: Lines 224-225. We have a different types of viscosity. Which one have you measured?
Response 1: We appreciate the reviewer’s attention to this important detail. In our study, we measured the apparent viscosity of the drug-loaded chitosan hydrogels using a digital viscometer. This instrument operates based on rotational shear, and as the hydrogels are non-Newtonian in nature, the measured apparent viscosity reflects the fluid’s resistance to flow under specific shear conditions. We modified the manuscript in line 226 to emphasize this point.
Comment 2: Why did you employ the Hixon-Crowell model? This expression applies to dosage forms such as tablets. It is appropriate to describe the release profile keeping in mind the diminishing surface of the drug particles during the dissolution. I suggest to use the Korsmeyer-Peppas equation, that, apart from the release rate constant, gives the n value describing the release mechanism.
Response 2: Thank you for pointing this out. The Hixson-Crowell model is commonly applied to drug delivery systems where the release process is associated with changes in the surface area and dimensions of the dosage form, such as erosion, degradation, or dissolution of the matrix. In our study, the amikacin-coated implant surface exhibited structural changes during the release process, as the polymeric matrix undergoes gradual degradation or erosion. Therefore, the Hixson-Crowell model was selected to describe the release kinetics, as it provides a theoretical basis for systems where drug release is controlled not only by diffusion but also by the reduction of surface area over time.
We agree that the Korsmeyer-Peppas model is used widely for polymer-based drug delivery systems. We analyzed the Korsmeyer-Peppas model for release mechanism and modified the manuscript in Table 2 (lines 550-551). Also, the release mechanism results were updated in lines 536-540 and lines 729-731 for discussion.
Comment 3: In line 284 there was information that you used also the second-order kinetics. However, I do not see the results from this model in Table 2.
Response 3: Thanks for your comment. We revised the writing error in the method section of the manuscript (lines 281-282).
Comment 4: Table 2: R2 is not a kinetic parameter. Please add the kinetic parameter such as the release rate constant for all models used.
Response 4: Agree. Table 2 was modified with the rate constants and values of “n” (lines 550-551).
Comment 5: Can you present an example of the fitting of the release experimental data to the theoretical models?
Response 5: Thank you for your valuable suggestion. To address this, we included in the revised manuscript a representative example figure illustrating the fitting of the experimental drug release data to theoretical models with updated Figure 9D (line 542).
Reviewer 3 Report
Comments and Suggestions for Authors
The authors investigated the use of amikacin (AMK), a broad spectrum aminoglycoside antibiotic, incorporated onto 3D-printed 316L stainless steel implants using biodegradable polymer coatings of chitosan and poly lactic-co-glycolic acid (PLGA). Low molecular weight chitosan (2%, w/v) and medium molecular weight chitosan (2%, w/v) were dissolved in a 1% (v/v) acetic acid solution was used. Before coating, all metal specimens were autoclaved and exposed to UV radiation for sterilization. All implant coating processes were performed under sterile conditions with UV exposure in the printer. The surface morphology was analyzed using microscopy and roughness and strength of the material was also evaluated. Further, the in vitro drug release profiles were assessed by incubating drug-loaded implants at 37 °C for one month keeping them immersed in PBS by a standardized method by the authors done earlier using HPLC. Further, the antimicrobial activity using various microorganisms were assessed by MIC and zone of inhibition assay. The study is very extensive and the authors have conducted many experiments supporting their hypothesis. I have a few technical comments:
1. Since chitosan is dissolved in acetic acid, its pH gets dropped. Did the authors check the pH of the chitosan solution before coating it? If not, why?
2. The drug release was calculated for 1 month. Did the authors check the biocompatibility of the released drug or polymers using cell lines or hemolysis assay? This is very important when we propose a new implant material to avoid post surgery infection. Please perform.
I recommend a major revision.
Author Response
The authors investigated the use of amikacin (AMK), a broad spectrum aminoglycoside antibiotic, incorporated onto 3D-printed 316L stainless steel implants using biodegradable polymer coatings of chitosan and poly lactic-co-glycolic acid (PLGA). Low molecular weight chitosan (2%, w/v) and medium molecular weight chitosan (2%, w/v) were dissolved in a 1% (v/v) acetic acid solution was used. Before coating, all metal specimens were autoclaved and exposed to UV radiation for sterilization. All implant coating processes were performed under sterile conditions with UV exposure in the printer. The surface morphology was analyzed using microscopy and roughness and strength of the material was also evaluated. Further, the in vitro drug release profiles were assessed by incubating drug-loaded implants at 37 °C for one month keeping them immersed in PBS by a standardized method by the authors done earlier using HPLC. Further, the antimicrobial activity using various microorganisms were assessed by MIC and zone of inhibition assay. The study is very extensive and the authors have conducted many experiments supporting their hypothesis. I have a few technical comments:
Thank you very much for taking the time to review this manuscript. We appreciate your valuable suggestions and careful evaluation of our work. Please find the detailed responses below and the corresponding revisions in track changes in the resubmitted files.
Comment 1: Since chitosan is dissolved in acetic acid, its pH gets dropped. Did the authors check the pH of the chitosan solution before coating it? If not, why?
Response 1: Thank you for pointing this out. We measured the pH of 1% (v/v) acetic acid solution, and it was around 2.9. We modified the manuscript in line 180 to emphasize this point.
Comment 2: The drug release was calculated for 1 month. Did the authors check the biocompatibility of the released drug or polymers using cell lines or hemolysis assay? This is very important when we propose a new implant material to avoid post surgery infection. Please perform.
Response 2: Thank you for your insightful comment. We appreciate the emphasis on biocompatibility, which is indeed critical when proposing new implant materials. The polymers used in our coating (PLGA and chitosan) have well-established safety profiles. PLGA has been approved by the US FDA in over 20 drug products, and chitosan-based formulations are currently undergoing multiple human clinical trials, as supported by the references provided below.
While we did not perform in vitro biocompatibility assays such as hemolysis or cell viability tests in this study, we acknowledge their importance. Our future research is directed to include comprehensive biocompatibility assessments, including cytotoxicity and hemocompatibility assays, as well as in vivo evaluations in animal models (e.g., horses and alpacas) to assess both safety and osteointegration.
We have added the following statement in the revised manuscript: Our future research is directed to include comprehensive biocompatibility assessments, including cytotoxicity and hemocompatibility assays, as well as in vivo evaluations in animal models (e.g., horses and alpacas) to assess both safety and osteointegration.
Kantak MN, Bharate SS. Analysis of clinical trials on biomaterial and therapeutic applications of chitosan: A review. Carbohydrate Polymers. 2022 Feb 15;278:118999. Notario-Pérez F, Martín-Illana A, Cazorla-Luna R, Ruiz-Caro R, Veiga MD. Applications of chitosan in surgical and post-surgical materials. Marine drugs. 2022 Jun 15;20(6):396.
Teixeira-Santos R, Lima M, Gomes LC, Mergulhão FJ. Antimicrobial coatings based on chitosan to prevent implant-associated infections: A systematic review. Iscience. 2021 Dec 17;24(12).
Wang Y, Qin B, Xia G, Choi SH. FDA’s poly (lactic-co-glycolic acid) research program and regulatory outcomes. The AAPS Journal. 2021 Jun 29;23(4):92.
Gentile P, Chiono V, Carmagnola I, Hatton PV. An overview of poly (lactic-co-glycolic) acid (PLGA)-based biomaterials for bone tissue engineering. International journal of molecular sciences. 2014 Feb 28;15(3):3640-59.
Omidian H, Wilson RL. PLGA Implants for Controlled Drug Delivery and Regenerative Medicine: Advances, Challenges, and Clinical Potential. Pharmaceuticals. 2025 Apr 27;18(5):631.
Round 2
Reviewer 3 Report
Comments and Suggestions for Authors
The authors have addressed all the queries raised by honorable reviewers and the manuscript is now much improved. Although, the biocompatibility assay that I suggested was not done. The authors have given some justification for not doing the experiement which is satisfactory. I accept the article in its present form for publication.